# OpenReview forum: "MACPO: Weak-to-Strong Alignment via Multi-Agent Contrastive Preference Optimization"
_ICLR.cc/2025/Conference — ICLR 2025 Poster_

### Official Review · Reviewer_ffho · 2024-10-31

**Soundness:** 3
**Presentation:** 1
**Contribution:** 3
**Rating:** 6
**Confidence:** 4

**Summary:**

The paper proposes Multi-Agent Contrastive Preference Optimization (MACPO), which aims at letting strong students and weak teachers to learn from each other by encouraging unfamiliar positive behaviors and penalizing familiar negative behaviors. The proposed algorithm achieves better performance compared with other weak-to-strong alignment methods on helpfulness and harmlessness benchmarks.

**Strengths:**

1. The proposed approach is intuitive and partially solves the problem of collapsing by learning through self-generated data.

2. The ablation study is comprehensive, validating the claims. The authors clearly illustrate the benefits brought up by unfamiliar positive behavior and familiar negative behavior.

**Weaknesses:**

1. The writing can be significantly improved, especially for section 4. The notation often comes with 4 to 5 super and subscripts, which is very difficult to follow. I highly recommend the authors clean this part up.

2. The proposal is intuitive, but the concept of “familiar” and “unfamiliar” is not well defined or discussed. Does familiar mean self-generated content and unfamiliar mean content generated by other models? What are the measures you can use for determining familiarity?

**Questions:**

If I understand correctly, when constructing positive behaviors, one selection criteria is that the weak labels should have low perplexity when evaluated on the strong student. However, doesn’t this low perplexity mean that this weak label is familiar to the strong student? Because low perplexity indicates that this weak label is likely to be generated by the strong student as well. This seems to contradict the goal of generating unfamiliar positive behavior.

---

> ### Author Response · Authors · 2024-11-22
> **Response to reviewer ffho**
>
> We would like to thank the reviewer for the thorough review, which will help to improve our paper. Our reply can be summarized as the following:
>
> **1. Regarding notations**
>
> Thanks for your suggestions. To make the method easy to follow, we have simplified notations of weak teacher and strong student models in Section 4 of the rebuttal revision paper.
>
> **2. The definitions of  “familiar” and “unfamiliar”**
>
> Thanks for your feedback. Since we assume different LLMs have different knowledge, familiar behaviors refer to self-generated samples, whereas unfamiliar behaviors refer to samples generated by other models. We have included these definitions in Sections 1 and 4 of the rebuttal revision paper.
>
> **3.  Regarding positive behavior construction**
>
> Thanks for your detailed review and feedback.  Our key idea of positive behavior construction is to select unfamiliar and high-quality labels for training the student model. To achieve this, we first identify unfamiliar labels as those generated by teacher models, rather than self-generated labels. Then, we use low perplexity to further filter high-quality labels among these unfamiliar ones.  While higher perplexity does indicate greater unfamiliarity, it often corresponds to lower-quality labels due to increased noise.
> To clarify this motivation, we conduct an experiment using helpfulness questions. Specifically, we randomly sample 100 labels generated by teacher and student models, and then calculate the perplexity and reward for these labels. As shown in the table below, labels generated by teachers are categorized as unfamiliar based on the perplexity of the student model. Among these unfamiliar labels, the highest-quality ones are those generated by the 8B Llama3 teacher, which exhibit the lowest perplexity and the highest reward. Conversely, labels generated by the 7B Llama2 teacher have the highest perplexity but the lowest reward. These results have been included in Appendix G.4 of the rebuttal revision paper.
>
> |                    | Perplexity of 70b Llama2 student | Reward |
> |--------------------|:--------------------------------:|:------:|
> | 70b Llama2 student |               9.80               |  37.96 |
> |  8b Llama3 teacher |               11.87              |  42.94 |
> | 7b Mistral teacher |               11.95              |  42.63 |
> |  7b Llama2 teacher |               12.00              |  42.31 |

---

> > ### Author Response · Authors · 2024-11-25
> > **Looking Forward to Your Reply**
> >
> > Dear Reviewer ffho,
> >
> > Thank you for the time and effort you have dedicated to reviewing our submission. We also appreciate your recognition of the key aspects of our work: proposing an intuitive approach, solving the collapsing problem, and demonstrating a comprehensive ablation study.
> >
> > We hope we have addressed the concerns raised in your initial reviews and eagerly await your thoughts and further guidance to refine our work. As the author-reviewer discussion period for ICLR 2025 will be over soon, please let us know if you require any additional information or clarification from our end. We are open to engaging in further discussions to enhance our submission. Thank you!

---

> > ### Comment · Reviewer_ffho · 2024-11-27
> >
> > Thank the authors for the reply and the addition of new experiments. I think the clarity improves. For the third point, I think "While higher perplexity does indicate greater unfamiliarity, it often corresponds to lower-quality labels due to increased noise." is kind of intuitive, but might require more evidence. This is also an important motivation for the algorithmic design. Thus, in the final manuscript, I suggest the authors to clearly state this claim and pay more attention on validating it. I will retain my positive score towards acceptance.

---

> ### Author Response · Authors · 2024-11-27
> **Response to reviewer ffho (Part 2)**
>
> Thank you again for your careful review and feedback, and we are glad that the clarity has improved. Our reply can be summarized as the following:
>
> **1. Regarding perplexity-based filtering technique**
>
> Thanks for the feedback.
>
> **For relevant reference papers**, previous works [1-3] have experimentally demonstrated the effectiveness of filtering high-quality data using low perplexity. Based on these findings, we use perplexity to filter high-quality data generated by multiple weak teachers. We have cited these relevant reference papers in Section 4.1 of the rebuttal revision paper.
>
> **For the experiment evidence**, as shown in Figure 2, we find that the performance of strong students increases as the number of weak teachers increases at each iteration. This observation indirectly supports the use of low perplexity in selecting higher-quality positive samples from multiple weak teachers.
>
> **To further address concerns**, we replace perplexity filtering with random sampling to directly assess the effectiveness of perplexity filtering. As shown in the table below, we observe that removing the perplexity filtering of weak labels (-ppl filtering) decreases the performance of helpfulness and harmlessness. This demonstrates that perplexity filtering is effective for filtering high-quality samples for helpfulness and harmlessness. We have included these results in Appendix G.2 of the rebuttal revision paper.
>
> |                        | HH-Helpful | HH-Harmless | PKU-SafeRLHF |  Average  |
> |------------------------|:----------:|:-----------:|:------------:|:---------:|
> |      MACPO (iter1)     |    58.06   |    59.20    |     61.16    |   59.47   |
> |      MACPO (iter2)     |    69.08   |    69.55    |     63.43    |   67.35   |
> |      MACPO (iter3)     |  **69.81** |  **70.25**  |   **63.49**  | **67.85** |
> | -ppl filtering (iter1) |    49.05   |    59.16    |     57.85    |   55.35   |
> | -ppl filtering (iter2) |    67.74   |    62.96    |     63.18    |   64.63   |
> | -ppl filtering (iter3) |    67.89   |    62.49    |     63.12    |   64.50   |
>
> [1] When Less is More: Investigating Data Pruning for Pretraining LLMs at Scale. NIPS@ATTRIB 2023.
>
> [2] Scaling data-constrained language models. NIPS 2023.
>
> [3] CCNet: Extracting high quality monolingual datasets from web crawl data. LREC 2019.
>
> We hope this can help address your concerns; please also let us know if you have anything else to ask.

---

### Official Review · Reviewer_Wdjd · 2024-11-03

**Soundness:** 2
**Presentation:** 3
**Contribution:** 2
**Rating:** 6
**Confidence:** 5

**Summary:**

This paper introduces MACPO, a framework for weak-to-strong alignment that enables strong language models to learn from weaker teachers. The authors claim that MACPO enhances positive behavior exchange and alignment refinement through iterative strategies, improving both teacher and student alignment, and results on benchmark datasets show MACPO’s effectiveness, with stronger alignment as the number of weak teachers grows.

**Strengths:**

The authors clearly define the problem, making it easy to understand the motivations behind their approach. This clarity in the problem statement helps to set a strong foundation for the rest of the work. And this paper opens up several avenues for future research, providing a solid foundation for follow-up studies. The authors discuss limitations and possible extensions, showing an awareness of the field's current needs and future directions.

**Weaknesses:**

The most concerning aspect is the statement in line 224: *“Since a high perplexity ppl of the positive strong student indicates weak labels may contain negative noises,”* which appears to form the foundational assumption of the entire framework. I have not encountered a statement like this before. In my opinion, alignment and preference learning should prioritize evaluation metrics as the basis for setting up metrics. Providing substantial evidence, such as a reference paper or experiments to demonstrate this metric’s validity and necessity, would be essential for this work. However, I could not find corresponding evidence, which may represent a fundamental weakness. At least the author needs to show a positive correlation between perplexity and answer quality (although this positive correlation is also very weak evidence to me, considering that correlation does not mean causation). However, the author did not even show the most basic evidence, which is very limited.

I outline other concerns:

1. I find the pipeline somewhat confusing. According to Algorithm 1, the proposed method first derives the “Strong Student” before proceeding with “Weak Teacher” training. Typically, in teacher-student learning paradigms, the teacher model is trained first and then provides supervision for the student model. If my understanding of the pipeline is correct, could the authors clarify the rationale behind this approach?

2. It appears that the model sizes listed in Table 1 are inconsistent. For example, MACPO results are reported using both the 8B and 70B models, while other approaches only rely on the 8B model. This discrepancy may undermine the fairness of the comparisons.

3. In Table 1, how are the final evaluation results derived based on a third-party reward model? With multiple gold models available in RewardBench, it would be helpful for the authors to explain their choice of this particular model over others.

4. For HH-RLHF training, it’s unclear why the authors opted to use only 10K samples, which represent less than 10% of the original dataset. Do the final results depend heavily on this specific subset? This choice should be validated, as it raises concerns about the generalizability of the experimental results.

5. Are the experiments restricted solely to alignment for helpfulness and harmlessness? It would be beneficial to extend these experiments to other tasks, such as Reddit TL; DR. Even if the approach does not perform well on such tasks, presenting these results could offer valuable insights.

**Questions:**

Please refer to the weakness, I'm happy to modify my rate based on the response of the authors and refer to other reviewers' comments.

---

> ### Author Response · Authors · 2024-11-22
> **Response to reviewer Wdjd (Part one)**
>
> We thank the reviewer for the very detailed comments, our reply can be summarized as the following:
>
> **1. Regarding perplexity-based filtering technique**
>
> Thanks for the feedback.
>
> **For relevant reference papers**, previous works [1-3] have experimentally demonstrated the effectiveness of filtering high-quality data using low perplexity. Based on these findings, we use perplexity to filter high-quality data generated by multiple weak teachers. We have cited these relevant reference papers in Section 4.1 of the rebuttal revision paper.
>
> **For the experiment evidence**, as shown in Figure 2, we find that the performance of strong students increases as the number of weak teachers increases at each iteration. This observation indirectly supports the use of low perplexity in selecting higher-quality positive samples from multiple weak teachers.
>
> **To further address concerns**, we replace perplexity filtering with random sampling to directly assess the effectiveness of perplexity filtering. As shown in the table below, we observe that removing the perplexity filtering of weak labels (-ppl filtering) decreases the performance of helpfulness and harmlessness. This demonstrates that perplexity filtering is effective for filtering high-quality samples for helpfulness and harmlessness. We have included these results in Appendix G.2 of the rebuttal revision paper.
>
> |                        | HH-Helpful | HH-Harmless | PKU-SafeRLHF |  Average  |
> |------------------------|:----------:|:-----------:|:------------:|:---------:|
> |      MACPO (iter1)     |    58.06   |    59.20    |     61.16    |   59.47   |
> |      MACPO (iter2)     |    69.08   |    69.55    |     63.43    |   67.35   |
> |      MACPO (iter3)     |  **69.81** |  **70.25**  |   **63.49**  | **67.85** |
> | -ppl filtering (iter1) |    49.05   |    59.16    |     57.85    |   55.35   |
> | -ppl filtering (iter2) |    67.74   |    62.96    |     63.18    |   64.63   |
> | -ppl filtering (iter3) |    67.89   |    62.49    |     63.12    |   64.50   |
>
>
> **2. Regarding the training order of models in algorithm**
>
> Thank you for your detailed feedback. The algorithm consists of two stages: initialization (line 974-976) and iterative optimization (line 977-996). **For the initialization stage**, the teacher model is trained first using golden labels. Then, the initialized teacher model provides supervision to initialize the student model. **After that, for the iterative optimization stage**, we iteratively optimize the student model and then optimize the teacher model.
>
> This reason is that we find that the initialized student model is not well aligned with the teacher model, so we further optimize the student model to improve the alignment performance, and iteratively optimize the teacher and the student then. To address potential confusion, we have updated Algorithm 1 to clearly separate the initialization stage from the iterative optimization stage. Additionally, we have added more detailed explanations in Appendix A of the rebuttal revision paper.
>
> **3.  Regarding the comparison setting in Table 1**
>
> Thank you for your feedback. Table 1 contains strong-to-weak alignment, self-alignment, and weak-to-strong alignment baselines. Different methods train the same 70b student model using different training strategies. Finally, we consistently evaluate the performance of 70b student models trained with different methods.
>
> **4.  Regarding the third-party reward model**
>
> Thanks for the feedback.
>
> **For the specific use of the third-party reward model**, following the previous paper [4], we concatenate the question and the model's answer as inputs to the reward model. The reward model's output is then scaled to a range of 0–1 using the sigmoid function. Finally, to facilitate comparison, we further scaled scores to 0–100 by multiplying by 100. We have updated the details of the reward calculation in Appendix D.1 of the rebuttal revision paper.
>
> **For the choice of the third-party reward model**, as we discussed in section 5.4, we follow the previous paper [4] to choose the oasst-rm-2-pythia-6.9b-epoch-1 [5] as our third-party reward model, for the fair comparison. Additionally, Table 10 in the RewardBench paper [6] demonstrates the effectiveness of this reward model, which achieves 92.5 out of 100 in evaluating alignment performance for conversational tasks. Furthermore, as shown in Tables 2 and 3, GPT-4 and human evaluations provide results consistent with those of the reward model, confirming its reliability.

---

> > ### Author Response · Authors · 2024-11-22
> > **Response to reviewer Wdjd (Part two)**
> >
> > **5.  Regarding the number of training samples**
> >
> > Thanks for your feedback. In the weak-to-strong alignment tasks, we need to iteratively train multiple 70b models for diverse baselines. Due to the limitations of computational resources in academia, we randomly sample a sufficient number of samples in datasets, to conduct the experiments. Specifically, to ensure that the number of samples is sufficient to train LLMs, we follow the previous works Alpaca [7] and Vicuna [8] by choosing the same order of magnitude of 10k samples for training. Moreover, we conduct extensive experiments on multiple datasets, including HH-RLHF and PKU-SafeRLHF. Experimental results on different datasets consistently demonstrate the effectiveness of our method.
> >
> >
> > **6. Evaluation on other alignment  tasks**
> >
> > Thanks for the suggestion.  Since we focus on aligning LLMs with helpfulness and harmlessness in weak-to-strong learning settings, we have conducted experiments on HH-RLHF and PKU-SafeRLHF. Experimental results on these datasets consistently demonstrate the effectiveness of our method. To further validate the performance of our method on general alignment tasks. We conduct experiments on the MT-Bench dataset [9]. This dataset encompasses a diverse range of tasks, including writing, roleplay, reasoning, math, coding, extraction, STEM, and humanities questions. Following previous work [9], we use the GPT-4 to evaluate the model output with scores ranging from 1-10. Since MT-Bench contains general questions for assessing helpfulness, we directly evaluated methods trained on helpfulness datasets without additional fine-tuning. For self-alignment methods and MACPO, we choose checkpoints with the highest rewards in Table 1 for evaluation. As illustrated in the table below, our method, MACPO, consistently outperforms the baselines. Furthermore, these results illustrate the ability of our method to generalize to other alignment tasks. We have included these results in Appendix G.3 of the rebuttal revision paper.
> >
> >
> > |     Method     | MT-Bench |
> > |:--------------:|:--------:|
> > |     RLAIF      |   4.16   |
> > |      RLCD      |   4.59   |
> > |      SPIN      |   2.56   |
> > | Self-rewarding |   3.69   |
> > |    Naive SFT   |   2.11   |
> > | Confident loss |   2.23   |
> > |      MACPO     | **4.63** |
> >
> >
> >
> >
> >
> > [1] When Less is More: Investigating Data Pruning for Pretraining LLMs at Scale. NIPS@ATTRIB 2023.
> >
> > [2] Scaling data-constrained language models. NIPS 2023.
> >
> > [3] CCNet: Extracting high quality monolingual datasets from web crawl data. LREC 2019.
> >
> > [4] Preference ranking optimization for human alignment. AAAI 2024.
> >
> > [5] https://huggingface.co/OpenAssistant/oasst-rm-2-pythia-6.9b-epoch-1
> >
> > [6] Rewardbench: Evaluating reward models for language modeling. 2024.
> >
> > [7] https://github.com/tatsu-lab/stanford_alpaca#data-release.
> >
> > [8] Instruction tuning with gpt-4. 2023.
> >
> > [9] Judging LLM-as-a-Judge with MT-Bench and Chatbot Arena. NIPS 23.

---

> ### Author Response · Authors · 2024-11-25
> **Looking Forward to Your Reply**
>
> Dear Reviewer Wdjd,
>
> Thank you for dedicating your time and effort to reviewing our submission. We sincerely appreciate your recognition of the key aspects of our work: clearly defining the problem, effectively demonstrating the motivations, and providing a solid foundation for future studies.
>
> We hope we have addressed the concerns raised in your initial reviews and eagerly await your thoughts and further guidance to refine our work. As the author-reviewer discussion period for ICLR 2025 will be over soon, please let us know if you require any additional information or clarification from our end. We are open to engaging in further discussions to enhance our submission. Thank you!

---

> > ### Author Response · Authors · 2024-11-28
> > **Awaiting Your Feedback**
> >
> > Dear Reviewer Wdjd,
> >
> > Thank you again for reviewing our submission. As the author-reviewer discussion period for ICLR 2025 is nearly over, please let us know if any further information or clarification is needed. We are ready to engage in any further discussions with you！ Looking forward to your further feedback!

---

> > > ### Comment · Reviewer_Wdjd · 2024-11-28
> > >
> > > Thanks for the reviewers' response and additional experiments. My concern is well addressed. I will raise my score.e

---

> > > > ### Author Response · Authors · 2024-11-28
> > > >
> > > > Thanks for your recognition and constructive suggestions, which have been instrumental in enhancing the quality of our research!

---

### Official Review · Reviewer_Dk7P · 2024-11-04

**Soundness:** 3
**Presentation:** 3
**Contribution:** 3
**Rating:** 8
**Confidence:** 4

**Summary:**

This paper proposes a multi-agent contrastive preference optimization (MACPO) approach for weak-to-strong alignment in LLMs. The authors utilize a multi-agent framework that iteratively improves both weak teachers and strong students by reinforcing unfamiliar positive behaviors and penalizing familiar negative ones. The experimental results show that MACPO achieves enhanced alignment performance over traditional methods, particularly as the number of weak teachers increases.

The main contributions of the paper include: 1) introducing the MACPO framework for weak-to-strong alignment; 2) incorporating mutual positive behavior augmentation and hard negative behavior construction to support iterative improvements; and 3) validating the proposed method's effectiveness through experiments on helpfulness and harmlessness alignment datasets​.

**Strengths:**

1. The idea of using a multi-agent contrastive preference optimization approach to achieve weak-to-strong alignment is innovative. Most existing work focuses on direct fine-tuning or reinforcement learning methods to improve alignment, but these approaches cannot leverage the incremental learning power of weak teachers to iteratively refine alignment, limiting the model’s ability to generalize across various behaviors. This paper introduces a method that enhances alignment by reinforcing positive unfamiliar behaviors and penalizing negative familiar ones, as well as significantly improves alignment performance as more weak teachers contribute to training.

2. The workflow is well-structured, as it combines contrastive preference optimization with a multi-agent setup to make the alignment process adaptable and iterative. This approach encourages behavior diversity among agents and further enhances the robustness and effectiveness of the alignment process.

3. The experiments are extensive, with detailed analysis of the results. These experiments validate the effectiveness of the MACPO framework and demonstrate the scalability and adaptability of the method across models with different sizes and complexities​.

**Weaknesses:**

1. The experiments use only two alignment datasets, expanding the evaluation to include additional alignment tasks like toxicity detection or complex reasoning would provide a more comprehensive assessment of MACPO's generalizability.

2. In Section 4.2 HARD NEGATIVE BEHAVIOR CONSTRUCTION, the paper does not clearly explain how agent interactions are managed or how behaviors are tracked across weak and strong agents throughout the alignment process.

**Questions:**

In Section 3.1, the paper assumes that using weak teachers can incrementally improve alignment by reinforcing positive behaviors. Could the authors clarify what criteria define a "weak teacher" and how its effectiveness in improving alignment is measured compared to using strong agents from the beginning?

The paper describes using multiple agents for reinforcement but lacks details on whether adjustments are made dynamically based on each agent's performance. Are there mechanisms in place to adjust reinforcement strategies depending on agent success rates, and if so, how are these adjustments implemented to ensure optimal alignment?

---

> ### Author Response · Authors · 2024-11-22
> **Response to reviewer Dk7P**
>
> We sincerely thank the reviewer for valuable suggestions. These contributions are instrumental in enhancing the quality of our work. Below is a summary of our responses to your comments:
>
> **1. Evaluation on other alignment  tasks**
>
> Thanks for the suggestion.  Since we focus on aligning LLMs with helpfulness and harmlessness in weak-to-strong learning settings, we have conducted experiments on HH-RLHF and PKU-SafeRLHF. Experimental results on these datasets consistently demonstrate the effectiveness of our method. To further validate the performance of our method on general alignment tasks. We conduct experiments on the MT-Bench dataset [1]. This dataset encompasses a diverse range of tasks, including writing, roleplay, reasoning, math, coding, extraction, STEM, and humanities questions. Following previous work [1], we use the GPT-4 to evaluate the model output with scores ranging from 1-10. Since MT-Bench contains general questions for assessing helpfulness, we directly evaluated methods trained on helpfulness datasets without additional fine-tuning. For self-alignment methods and MACPO, we choose checkpoints with the highest rewards in Table 1 for evaluation. As illustrated in the table below, our method, MACPO, consistently outperforms the baselines. Furthermore, these results illustrate the ability of our method to generalize to other alignment tasks. We have included these results in Appendix G.3 of the rebuttal revision paper.
>
> |     Method     | MT-Bench |
> |:--------------:|:--------:|
> |     RLAIF      |   4.16   |
> |      RLCD      |   4.59   |
> |      SPIN      |   2.56   |
> | Self-rewarding |   3.69   |
> |    Naive SFT   |   2.11   |
> | Confident loss |   2.23   |
> |      MACPO     | **4.63** |
>
>
>
> **2. Details of negative agents**
>
> Thanks for your feedback. As discussed in section 4.3, during our iteration optimization process, these negative agents are frozen, to consistently provide familiar negative behavior and avoid the degradation of the positive agents.
>
> [1] Judging LLM-as-a-Judge with MT-Bench and Chatbot Arena. NIPS 23.

---

> > ### Author Response · Authors · 2024-11-25
> > **Looking Forward to Your Reply**
> >
> > Dear Reviewer Dk7P,
> >
> > Thank you for dedicating your time and effort to reviewing our submission. We also appreciate your recognition of the three key aspects of our work: proposing an innovative idea, presenting a well-structured approach, and demonstrating extensive experimental results and detailed analysis.
> >
> > We hope we have addressed the concerns raised in your initial reviews and eagerly await your thoughts and further guidance to refine our work. As the author-reviewer discussion period for ICLR 2025 will soon conclude, please let us know if you require any additional information or clarification. We are open to engaging in further discussions to enhance our submission.  Thank you!

---

> > > ### Comment · Reviewer_Dk7P · 2024-11-27
> > >
> > > Thanks for the follow-up and the new experiments. They addressed most of my concerns and thus I have raised my rating.

---

> > > > ### Author Response · Authors · 2024-11-27
> > > > **Response to reviewer Dk7P (Part 2)**
> > > >
> > > > Thank you for your support of our work. Your valuable feedback has made our work better!

---

### Official Review · Reviewer_sBPe · 2024-11-04

**Soundness:** 4
**Presentation:** 2
**Contribution:** 3
**Rating:** 8
**Confidence:** 4

**Summary:**

The paper explores the alignment problem when the LLM outperforms humans and human supervision is therefore weak. The authors propose a multiagent contrastive preference optimization (MACPO) framework to for weak-to-strong alignment by iteratively reinforcing unfamiliar positive behaviors while penalizing familiar negative ones. The authors evaluate their method on HH-RLHF and PKU-SafeRLHF datasets and show that MACPO improves alignment performance of strong students and weak teachers.

**Strengths:**

1. The research problem is important and timely, as large language models are rapidly advancing and achieving near-human capabilities, and allowing them to surpass human performance with potentially only weak human supervision is a critical issue.
2. The paper is well-written and clearly structured, with a good introduction and related work section, clearly described methodology, and well-organized experiments and results.
3. The experimental results are promising. The scale of the experiments, diversity of evaluation, and the rich comparison are appreciated.

**Weaknesses:**

1. While the description of the MACPO framework in section 4.1, 4.2 is clear, the introduction of the framework in line 053 and line 196 is confusing. This is mainly due to the use of term 'behavior' and 'familiar' before they are defined. Moreover, the paragraph in line 196 is almost the same as the paragraph in line 061, which does not provide improved clarity.
2. In Section 3.1, the formulation of the problem, the authors adopt an analogy setting where weak teachers are fine-tuned small models and strong students are big models 'initialized on weak labels generated by weak teachers for the held-out question set(line 261)'. In this case, it is not clear whether the strong student is truly strong, and it is not clear how this situation relates to the motivation problem in line 013.
3. Computation efficiency discussion and comparison would be helpful as the proposed method requires multiple iterations of optimizing multiple models. Would the Strong-to-weak alignment and Self-alignment benefit from the longer training time (with the same effective computation as MACPO)?

**Questions:**

Please see the weaknesses 2,3

---

> ### Author Response · Authors · 2024-11-22
> **Response to reviewer sBPe**
>
> We appreciate your time and effort in reviewing this paper and providing these insightful comments. We respond as follows:
>
>
> **1. Clarification of our method**
>
>
> Thank you for the feedback. To address your concerns, we have clarified the terms "familiar behaviors" and "unfamiliar behaviors" by defining them explicitly: familiar behaviors refer to self-generated samples, whereas unfamiliar behaviors refer to samples generated by other agents. These definitions have been added in both Sections 1 and 4 for better understanding. Additionally, to enhance the explanation of our method, we have expanded the details of the optimization stages in Section 4 of the rebuttal revision paper.
>
> **2. Weak-to-strong learning setting**
>
>
> To practically study the problem of humans supervising superhuman models, following the previous work [1], we adopt the simple analogical setup that uses weak models to supervise strong models. Weak and strong models are defined based on the strength of their base abilities before the fine-tuning stage. In the fine-tuning stage, strong students are restricted to training on labels generated by weak teachers. This design tests whether weak labels can effectively guide the behavior of strong students. In this paper, we use LLaMA2-7B-base (fewer parameters) as the weak teacher and LLaMA2-70B-base (more parameters) as the strong student. The LLaMA2 paper [2] demonstrates the significant gap in base abilities between the 7B and 70B models, supporting the rationale for this setup.
>
> **3. Time consumption and performance comparison**
>
> Thanks for the suggestion.
>
> **Time consumption comparison at each iteration.**
> After model initialization, compared to strong-to-weak alignment and self-alignment baselines, our method consumes similar time or less time at each iteration. Specifically, given the held-out question set, 7b LLMs take about 10 minutes to generate new samples, and the 70b LLMs take about 2 hours to generate new samples. For each iteration, MACPO requires sampling answers once from three 7b positive teacher models and one 70b positive student model, while RLAIF, RLCD and self-rewarding methods require sampling answers twice from 70b LLMs. Although SPIN only requires sampling once from 70b LLMs, it starts to decrease the alignment performance after the first iteration. We have included the computation efficiency discussion in Appendix E.2 of the rebuttal revision paper.
>
> **The iterative performance of strong-to-weak alignment methods.**
> We extend RLAIF and RLCD into iterative alignment methods by resampling samples at each iteration. As shown in the table below, MACPO consistently outperforms these methods across multiple iterations. The reason is that strong-to-weak alignment methods ignore further improving the teacher agents. We have added these results in Appendix G.1 of the rebuttal revision paper.
>
> |               | HH-Helpful | HH-Harmless | PKU-SafeRLHF |  Average  |
> |---------------|:----------:|:-----------:|:------------:|:---------:|
> | RLAIF (iter1) |    45.26   |    56.37    |     59.21    |   53.61   |
> | RLAIF (iter2) |    48.01   |    53.02    |     58.72    |   53.25   |
> | RLAIF (iter3) |    47.99   |    52.99    |    59.04     |   53.34   |
> |  RLCD (iter1) |    52.77   |    59.23    |     53.77    |   55.26   |
> |  RLCD (iter2) |    53.00   |    57.34    |     55.31    |   55.22   |
> |  RLCD (iter3) |    53.45   |    56.88    |     55.50    |   55.28   |
> | MACPO (iter1) |    58.06   |    59.20    |     61.16    |   59.47   |
> | MACPO (iter2) |    69.08   |    69.55    |     63.43    |   67.35   |
> | MACPO (iter3) |  **69.81** |  **70.25**  |   **63.49**  | **67.85** |
>
> **For the iterative performance of self-alignment methods**, as shown in Table 1, the performance of SPIN and Self-rewarding start to decrease after 1 or 2 iterations, which means that they have started to collapse. However, our approach can still further improve the performance of strong student LLM.
>
> [1] Weak-to-strong generalization: Eliciting strong capabilities with weak supervision. OPENAI 2023.
>
> [2] Llama 2: Open Foundation and Fine-Tuned Chat Models. Meta 2023.

---

> > ### Author Response · Authors · 2024-11-25
> > **Looking Forward to Your Reply**
> >
> > Dear Reviewer sBPe,
> >
> > Thank you for dedicating your time and effort to reviewing our submission. We also appreciate your recognition of the three key aspects of our work: addressing an important and timely problem, presenting a clearly structured approach, and demonstrating promising experimental results.
> >
> > We hope we have addressed the concerns raised in your initial reviews and eagerly await your thoughts and further guidance to refine our work. As the author-reviewer discussion period for ICLR 2025 will soon conclude, please let us know if you require any additional information or clarification. We are open to engaging in further discussions to enhance our submission. Thanks a lot!

---

> > ### Comment · Reviewer_sBPe · 2024-11-26
> > **Response to the authors**
> >
> > Thank you for the detailed explanations and results. Most of my concerns have been addressed, and I only have a few questions for further clarification.
> >
> > I understand that the supervisor models are 7B models while the model being updated is a 70B model, which aligns with prior work and is an interesting setup to me. Moreover, the pipeline appears promising for training large models in practice, on seeing the comparisons with RLAIF and RLCD.
> >
> > My question is more about, under the current experiment setting,  to what extent can the student model extend beyond the teacher model? This closely ties to the motivation you mentioned—"In scenarios where LLMs outperform humans, we face a weak-to-strong alignment problem." More specifically, since the positive signals during training come from best-of-k sampling by the teacher (based on perplexity), shall we expect the student to achieve better evaluation performance than the teacher with best-of-k sampling?
> >
> > Additionally, regarding computation, could you elaborate on the computational overhead involved in updating the three teacher models?

---

> > > ### Author Response · Authors · 2024-11-27
> > > **Response to reviewer sBPe (Part 2)**
> > >
> > > Thank you again for your helpful review, and we are glad that we addressed most of your concerns. We respond as follows:
> > >
> > > **1. Performance comparison between teachers with best-of-k sampling and students**
> > >
> > > Thanks for the feedback. We conduct experiments to evaluate teachers with best-of-k sampling and the student, presenting the results in the iterative optimization order. As shown in the table below, given weak supervision generated by three teachers using best-of-k sampling, the student demonstrates better evaluation performance than teachers with best-of-k sampling.
> > >
> > > |                      Model                     | HH-Helpful | HH-Harmless | PKU-SafeRLHF | Average  |
> > > |:----------------------------------------------:|:----------:|:-----------:|:------------:|:--------:|
> > > | 3 initialized teachers with best-of-k sampling |    50.35   |    56.22    |     51.71    |   52.76  |
> > > |                 Student (iter1)                |    58.06   |    59.20    |     61.16    |   59.47  |
> > > |   3 teachers with best-of-k sampling (iter 1)  |    66.20   |    62.91    |     61.92    |   63.68  |
> > > |                 Student (iter2)                |    69.08   |    69.55    |     63.43    |   67.35  |
> > > |   3 teachers with best-of-k sampling (iter 2)  |    69.53   |    69.96    |     63.24    |   67.58  |
> > > |                 Student (iter3)                |    69.81   |    70.25    |     63.49    |   67.85  |
> > > |   3 teachers with best-of-k sampling (iter 3)  |    70.33   |    70.38    |     64.63    |   68.45  |
> > >
> > >
> > > **2. More details of time consumption**
> > >
> > > Thanks for the feedback.
> > >
> > > For each iteration, MACPO involves sampling answers from three 7B positive teacher models and one 70B positive student model. Specifically, updating the student requires 30 minutes to sample positive answers from three teachers, while updating the three teachers takes 2 hours to sample positive answers from the student.
> > >
> > > In comparison, RLAIF, RLCD, and self-rewarding methods require 4 hours to sample answers twice from the student. Although SPIN requires only 2 hours to sample once from the student, its alignment performance starts to degrade after the first iteration.
> > >
> > > We hope this can help address your concerns; please also let us know if you have anything else to ask.

---

> > > > ### Comment · Reviewer_sBPe · 2024-11-27
> > > > **Response**
> > > >
> > > > The last set of experiments, together with those presented in part 1, are extremely insightful!
> > > >
> > > > Initially, I was confused by two main reasons.
> > > > - I wasn't convinced  that only unfamiliar/familiar behaviors had such a significant impact on  the weak model's ability to provide strong signals for training. What I understand and found more convinced from the responses is that essentially the authors leverage the efficiency and effectiveness of ensemble methods (best-of-k sampling) to enhance small models and enable mutual evolution of small and large models to keep improving the positive signals.
> > > > - I was initially misled by the descriptions of student and teacher as weak/strong models. This led me to assume an absolute strong-weak relation between models. Actually it can by very dynamically according to the status of teacher and student. Though I understand the term weak-to-strong alignment is more precise to follow the prior arts, I believe simple explanations, e.g. say it refers to the size/optimal capability of model, would be beneficial for a broader audience.
> > > >
> > > > I hope the authors consider the above points to further improve the paper. In light of my above understanding, I believe the paper  presents valuable insights to share at the conference. Thus I am willing to raise my score.

---

> > > > > ### Author Response · Authors · 2024-11-27
> > > > > **Response to reviewer sBPe (Part 3)**
> > > > >
> > > > > Thank you for increasing the score. We are glad that the last set of experiments experiment addresses your conerns.
> > > > >
> > > > > Your valuable suggestions greatly contribute to the quality of our manuscript. Thank you again for your precious time and valuable suggestions!

---

### Author Response · Authors · 2024-11-22
**General Response**

We sincerely thank all the reviewers for their insightful and valuable comments.

We appreciate the reviewers for agreeing that this paper clearly presents an innovative and intuitive idea for solving an important research problem weak-to-strong alignment. In particular, we propose a multi-agent framework that iteratively enhances both weak teachers and strong students by reinforcing unfamiliar positive behaviors while penalizing familiar negative ones. The experimental results demonstrate that MACPO outperforms existing baselines.

**And we thank the great work of all reviewers for recognizing the key aspects of our work:**

1. The motivation and novelty of MACPO are clear and reasonable. (All Reviewers)

2. The challenges in weak-to-strong alignment are clearly defined, and our solution is well-structured. (All Reviewers)

3. The experimental results are comprehensive and promising, effectively validating the effectiveness and adaptability of the MACPO framework. (Reviewers sBPe, Dk7P, and ffho)

**To address the reviewers’ concerns, we have provided additional details about our method and conducted several experiments:**

1. We explained more details about the method, experimental setting and evaluation.

2. We extended strong-to-weak alignment baselines into iterative methods to compare their iterative performance with MACPO.

3. We evaluated MACPO’s effectiveness on the MT-Bench, which includes diverse alignment tasks.

4. We performed a detailed ablation study to validate the effectiveness of the perplexity filtering technique in constructing positive behaviors.

5. We conducted experiments to further illustrate the motivation of the perplexity filtering technique.

Next, we address each reviewer’s specific concerns in detail.

---

### Meta-Review · Area_Chair_jRmo · 2024-12-19

**Metareview:**

The paper proposes Multi-Agent Contrastive Preference Optimization (MACPO) to encourage strong students and weak teachers to learn from each other through contrastive learning. All the reviewers found the paper easy to follow and are satisfied with the strong empirical results. During the initial reviews, there were some questions regarding the definition of certain terminology, such as "familiar" vs "unfamiliar" behavior to the LLMs, as well as the notation/presentation of the paper, which the authors successfully addressed and improved during rebuttal. All the reviewers are leaning towards acceptance and I encourage the authors to take the reviewers' comments in preparing the final camera-ready version of the paper.

Given the importance of the topic studied in this paper (as the current state-of-the-art models are growing larger and larger) and its extensive empirical results, I recommend for a spotlight presentation.

**Additional Comments On Reviewer Discussion:**

Reviewers actively engage with the authors during the rebuttal period and have reached a positive consensus towards acceptance.

---

### Decision · Program_Chairs · 2025-01-22

Accept (Poster)